# Toddler-Inspired Visual Object Learning

**Sven Bambach[1], David J. Crandall[1], Linda B. Smith[2], Chen Yu[2]**
[1]School of Informatics, Computing, and Engineering, [2]Dept. of Psychological and Brain Sciences
Indiana University Bloomington
{sbambach, djcran, smith4, chenyu}@iu.edu

## Abstract

Real-world learning systems have practical limitations on the quality and quantity of the training datasets that they can collect and consider. How should a system go about choosing a subset of the possible training examples that still allows for learning accurate, generalizable models? To help address this question, we draw inspiration from a highly efficient practical learning system: the human child. Using head-mounted cameras, eye gaze trackers, and a model of foveated vision, we collected first-person (egocentric) images that represent a highly accurate approximation of the "training data" that toddlers' visual systems collect in everyday, naturalistic learning contexts. We used state-of-the-art computer vision learning models (convolutional neural networks) to help characterize the structure of these data, and found that child data produce significantly better object models than egocentric data experienced by adults in exactly the same environment. By using the CNNs as a modeling tool to investigate the properties of the child data that may enable this rapid learning, we found that child data exhibit a unique combination of quality and diversity, with not only many similar large, high-quality object views but also a greater number and diversity of rare views. This novel methodology of analyzing the visual "training data" used by children may not only reveal insights to improve machine learning, but also may suggest new experimental tools to better understand infant learning in developmental psychology.

## 1  Introduction

Any learning system — human or machine — faces the challenge of building an accurate, general model of the world from a limited amount of training data. Both quality and quantity of the training data are critical to successful learning. In the field of computer vision, for example, much of the dramatic recent progress in object recognition accuracy has been due to massive new datasets like ImageNet [28]. While more training data is probably always better — very recent results show that recognition models continue to improve even as datasets reach into the *billions* of images [19] — both human and machine learning systems in the real world are bounded by practical constraints on the time and energy needed to collect and process a potentially infinite amount of training data.

The usual approach in machine learning is to simply collect and use as much data as possible, with the assumption that the training data is independently sampled from some underlying distribution that is representative of the examples encountered in the real world. Since quality and quantity of the training data are correlated, more quantity naturally leads to better overall quality. In the case of visual recognition of object categories, for example, large datasets like ImageNet [28] contain not only many instances of clear, canonical objects (e.g., side views of Toyota sedans) but also relatively few instances of many less common objects (e.g., oblique views of Model T's). The larger the dataset is, the more likely it contains high-quality data points (e.g., uncommon instances) for generalization.

In this paper our goal is to delve into the data side of machine learning, but we draw inspiration from perhaps the best known visual learning system — the human child. We know that by the age of

two, toddlers recognize instances of roughly 300 object categories [9]. They can also generalize a newly learned label to instances that they have never seen before [16]. Even though computational deep learning models trained with large databases of "natural" images have matched and sometimes outperformed humans' remarkable visual abilities [13], human toddlers are still more efficient learners: their visual learning relies on limited weakly-supervised training data from their individual experiences, while deep learning models can use much larger quantities of supervised training data. Recent studies in developmental and cognitive psychology suggest that successful learning in toddlers lies in the quality of visual data they collect from their everyday activities [7]. In particular, while most images in ImageNet and other computer vision datasets are photographs that were taken with consumer cameras by adults, the visual information cast on a toddler's retina is from a first-person perspective. The visual properties of images from the first-person and third-person views are very different [34]. Moreover, the visual experiences collected by toddlers are not a large collection of random pictures taken by many individuals in many locations and contexts, but are instead generated by a single person based on their everyday activities, and are thus much more coherent and correlated. We believe that the unique visual properties and distributions of imagery perceived by human toddlers are key to their success in visual object learning.

To test this idea, we used head-mounted cameras and eye trackers to record an approximation of the visual stimuli that infants receive while playing with toys in a naturalistic, everyday play environment. We use these data to train state-of-the-art object models — deep convolutional neural networks (CNNs) — on an object recognition task, and study the performance of the networks as we manipulate various properties of the training dataset. Our goal is *not* to build state-of-the-art object classifiers, nor is it to model the actual mechanism by which children learn; but to use CNNs to quantify and compare the information content of various datasets, i.e., to use them as *data mining* algorithms to measure which properties of visual data lead to better visual object learning. By doing so, we believe the methodology we propose here in the long-term may not only reveal ways of improving machine learning models by optimizing properties of training datasets, but also could lead to new computational tools to understand the underlying mechanisms behind infant learning.

## 2   Related work

A few recent studies have explored deep learning models in relation to data collected by human subjects in experimental contexts [3, 12, 23, 27], and have already led to important findings in both the machine and human learning fields. On one hand, experimental psychology approaches provide a framework to understand hidden computational principles and properties of sophisticated algorithms such as deep neural networks [27, 30, 32]. On the other hand, deep learning models can be a useful tool to reveal the representations and computations required for human learners to solve hard learning problems in the real world. Inspired by previous studies, the present study focuses on building

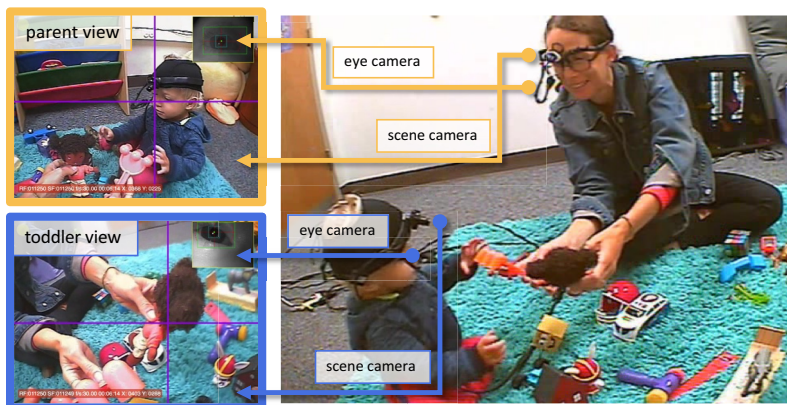

Figure 1: Our experimental setup. Child-parent dyads played together with a set of toys in a naturalistic environment, while each wore head-mounted cameras to collect egocentric video and eye gaze positions (left). A stationary camera recorded from a third-person perspective (right).

connections between visual object recognition in computer vision and toddler object learning in developmental psychology.

Our work connects with the data subset selection problem in machine learning, where the goal is to find a good subset of training examples [10, 11, 17, 21, 33, 37], often in the context of active learning [14, 20]. Most of this work formulates the problem mathematically for specific classifiers with specific objective functions. Our work is more generally related to the rich literature in data mining and machine learning on choosing subsets of data having certain properties. Of course, the most common approach to reducing the size of a dataset while preserving its statistical properties is random sampling. Other approaches include algorithms that represent a dataset with a few representative points [15, 24], try to correct for biases in training data [6], minimize or maximize the variation of data [4, 8, 25], sample from inliers while ignoring outliers [5], and so on. Instead of specific algorithms or mathematical formulations, we are interested in identifying more fundamental principles of what makes a good training dataset for visual object recognition and how children may naturally generate them in everyday contexts.

## 3 Methods

### 3.1 Data and Data collection

To closely approximate the training data used in toddler object learning, we collected visual data from everyday toy play — a context in which toddlers naturally learn about objects and their names. We used an experimental setup in which we placed a camera on a toddler's head to collect egocentric video of their field of view as they played with toy objects with their parent, as shown in Figure 1. We also used a head-mounted eye tracker to record their visual attention. These devices allowed us to record both egocentric video from the toddler's perspective as well as their gaze point on a moment-by-moment basis. We also collected video and gaze data from the parent in the same interaction. This experimental setup has been used successfully in studying infant perception and action [36], language learning [34], and social interaction [35]. Our study was reviewed and approved by the IRB at our institution.

Twenty-six child-parent dyads participated in our study. Each dyad was brought into a room with a set of 24 toys scattered on the floor; we used the same toys as in [3]. The children and parents were told to play with the toys, but no more specific directions were given. The children ranged in age from 15.2 to 24.2 months ($\mu$=19.4 months, $\sigma$=2.2 months). We collected five synchronized videos per dyad (head camera and eye camera for child, head camera and eye camera for parent, and a third person view camera – see Figure 1). The final dataset contains roughly 200 minutes of video, with each dyad contributing different amounts of data ranging from 3.4 minutes to 11.6 minutes ($\mu$=7.5 minutes, $\sigma$=2.3 minutes). The head-mounted cameras recorded video at 30 frames per second and $480 \times 640$ pixels per frame, with a horizontal field of view of about 70 degrees, and we used eye trackers from Positive Science [1]. We followed validated best practices for mounting the head cameras so as to best approximate subjects' actual first-person views, and to calibrate the eye trackers. The collected data were used to build different toy object training sets, as detailed in Sections 3.2 to 3.5.

We also used a separate image dataset (provided by [3]) of the same 24 toy objects captured by a third person camera in a controlled environment, some examples of which are shown in Figure 2c. The dataset consists of each object systematically photographed up-close against a black background from 128 viewpoints (different angles and distances), for a total of 3,072 images. These images are used as the test set for the experiments in Section 4.

### 3.2 Detecting Object Looks

From continuous gaze data, we manually coded, for each frame and each subject (toddlers and parents), which object the subject was attending to (if any). We defined an object look as either a toddler or parent continuously attending to the same object before switching their attention to another object. In total, there were 4,553 object looks for the toddlers and 5,052 object looks for the parents. The average object look duration was 1.86 seconds (56 frames) for toddlers and 1.40 seconds (42 frames) for parents. Some objects were attended more than others: the most popular toy attracted about four times as many looks as the least popular of the 24 toys. In this study, we used only the frames in which participants were looking at one of the 24 toys (e.g., we excluded looks at the other

person's face), totaling 258,250 frames for children and 217,290 frames for parents. The training data in our simulations (described below) consisted of the object instances attended to in these frames.

## 3.3 Detecting Objects

To avoid labor-intensive manual annotation, we used an automatic process to identify positions and sizes of objects visible in each frame. In particular, we used YOLO [26], a well-established detector that offers a good compromise between computational cost and accuracy. To train the model on our 24 objects, we manually annotated a small subset of about 1,200 frames (sampled from 15 randomly-chosen subjects) with bounding boxes. Although our models and YOLO detections are not perfect, the results are generally very good; for 97% of our (manually-annotated) child looks, YOLO correctly identified the attended object in at least one frame, while 83% of the parent looks had a correct YOLO recognition for at least one frame. The lower accuracy for parents was presumably because objects tend to be smaller in the parent's field of view (discussed below).

## 3.4 Simulating Acuity

Egocentric video captured by head-mounted cameras provides a good approximation of the field of view of the wearer. However, the human visual system exhibits well-defined contrast sensitivity due retinal eccentricity [22]: the area centered around the gaze point (the fovea) captures a high-resolution image, while the imagery in the periphery is captured at dramatically lower resolution due to its lesser sensitivity to higher spatial frequencies. As a result, the human visual system does not process all pixels equally in a first-person view image, but instead focuses more on the pixels around the fovea.

To closely approximate the visual signals that are "input" to a toddler's learning system, we simulated foveated vision in the egocentric view by applying a blurring function that preserves fine details only at the center of gaze. To do this, we started with the set of egocentric camera frames described above, where each frame is annotated with the $xy$-coordinate of the center of gaze as well as the class label of the attended object. We applied the method of Perry and Geisler [22] to simulate the effect of foveated visual acuity on each frame individually using their software implementation [2]. The basic idea is to preserve the original high-resolution image at the center of gaze, and increase blur progressively towards the periphery, as shown in Figures 2a and 2b. This technique applies a model of what is known about human visual acuity and has been validated with human psychophysical studies [22].

After gaze-based blurring to simulate acuity, we cropped out the area centered around the gaze coordinate to produce a training sample. We cropped out patches of different sizes, in order to simulate different amounts of visual data that may be processed by the toddler's learning system. In particular, we started with crops of $30°$ width (roughly corresponding to the area with high acuity) and increased the size in $10°$ increments up to $70°$ (the field of view of the head camera); examples are shown in Figures 2a and 2b. This procedure yields a total of 10 different datasets, corresponding to five different crop sizes ($30°$, ..., $70°$) for each of the two subject types (toddlers and parents).

## 3.5 Convolutional Neural Network models

In our experiments the goal is not to achieve the best possible accuracy using the latest recognition algorithm: we are interested in using machine learning classifiers to understand and characterize the properties of different training datasets, not to produce classifiers to be actually used for recognition. Our goal is also not to model the actual mechanism by which children learn: we are interested in characterizing the information and structure embedded in the data that *could* be learned by a good algorithm, human or computational. In other words, we use machine learning as a *data mining tool* that is useful for quantifying properties of visual data.

We use a well-known, state-of-the-art deep learning model, a convolutional neural network with the VGG16 [29] architecture, for image classification. Because our goal is not to optimize performance of the classifier itself, we use exactly the structure proposed in [29], except that we change the output layer to have 24 classes. We pre-trained the network on ImageNet [28], and used those weights to fine-tune the network on our training data, back-propagating across the whole network. Because the distribution of training examples (attended objects) is highly non-uniform, we used a categorical cross-entropy loss function that weighted the loss for each class to be inversely proportional to the

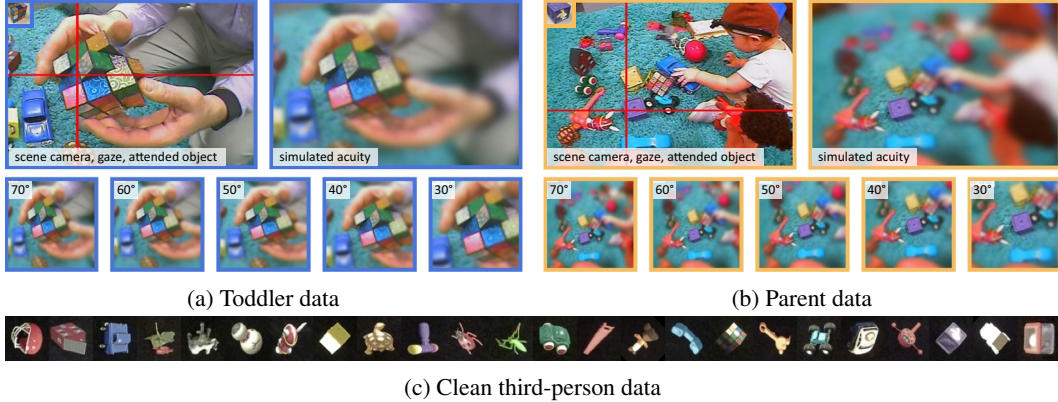

(a) Toddler data            (b) Parent data

(c) Clean third-person data

Figure 2: We use head-mounted cameras to collect egocentric video and eye gaze from parents and children. Moments of sustained attention on objects are annotated by human coders. (a,b): We simulate foveated vision using the eye gaze position, and generate training examples with varying amounts of context by cropping at different scales around the gaze point. (c): Clean images of all 24 toy objects in the experiment, randomly chosen from the test dataset.

number of training examples of that class,

$$\mathcal{L}(\theta) = -\sum_{i=1}^{N} \sum_{j=1}^{24} \frac{1}{N_j} \mathbb{1}(y_i = j) \ln f_j(x_i, \theta),$$

where $N$ is the total number of training examples, $N_j$ is the number of training examples for class $j$, $y_i$ is the correct class label for the $i$-th example, and $f_j(x_i, \theta)$ is the value of the output neuron corresponding to class $j$ when the network with weights $\theta$ is applied to example $x_i$. We used a standard stochastic gradient descent optimizer with a learning rate of $0.001$, momentum of $0.9$, and a batch size of 64 images. All training images were resized to $224 \times 224$ pixels, and we did not perform any data augmentation (e.g., left-right reflections or random croppings) since we wanted to use just the data that the infant learners receive.

Our goal is to quantify the information contained in a training dataset — i.e., how well an agent *could* learn to generalize given the training data — so we trained on the images derived from the head camera videos, but validated *and* tested our models on the "clean" object dataset described in Section 3.1 and Figure 2c. The motivation behind this is to train each first-person dataset to the point where it best generalizes to viewpoint-independent objects instances. We thus test each network on the clean dataset after every epoch and stop training once the accuracy has not increased for at least two epochs, and report the highest overall classification accuracy achieved up to that point. Because training is stochastic, we independently trained 10 networks for every condition and report the average classification accuracy as well as $95\%$ confidence intervals.

## 4 Experiments and Results

### 4.1 Quality of Toddler Data

We first compared the visual data perceived by toddlers with that perceived by their parents during the same interactions in the same environment. Figure 3a summarizes different views of one attended object (a red helmet), indicating a clear difference in object size and variation between the two groups. In light of this observation, we first quantified and compared two visual properties of object instances in the toddler's view with those in the parent's view.

First, as shown in Figure 3a, the distribution of object size was strikingly different across the two views: most instances of attended objects in the parent's view are smaller than 10% of the field of view, but objects in the toddler's view are much larger, with a large proportion of instances greater than 20% of the field of view. This may be due to several causes: children are shorter and closer to the ground, have shorter arms that make held objects appear larger, and may bring objects closer to

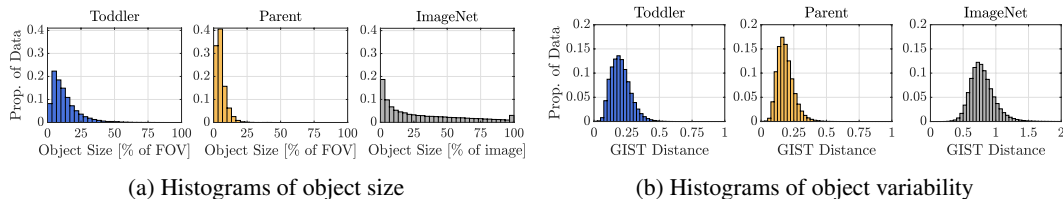

(a) Histograms of object size  (b) Histograms of object variability

Figure 3: Statistics of data collected by toddler and parent head cameras, with ImageNet as a baseline. (a): Distribution of training examples by object size (measured as the fraction of field of view). (b): Distribution of variation in the training examples (measured by distance between GIST features of pairs of examples of the same object/class).

their face than parents. But the effect is to create large objects in view (similar to ImageNet), which in turn create high-resolution images for learning — an idea we will test in Section 4.2.

Second, we compared visual similarity of object instances in order to quantify the diversity of the training data. In particular, we extracted GIST features [31] from each object instance and computed pairwise GIST distances (L2 norm) across all instances within each object category. We chose GIST because it is a low-level feature (as opposed to more semantic deep features) that is sensitive to the spatial orientation of an object; we wanted a distance metric such that two instances of the same object viewed from similar angles would have a small distance, while different views of the same object would have a large distance. As shown in Figure 3b, pairwise visual distances in parent data are rather small, indicating that object instances are similar to each other. For toddler data, there is also a portion of similar instances, but a bigger tail of instances that are not so similar to each other, showing variability within the training set. Thus, the combination of clustering and variability creates a unique distribution in toddler data which may benefit visual object learning — a conjecture that will be tested in Section 4.3. Note that the variability between objects in our data is expected to be much smaller than in ImageNet as we are comparing instances of the same physical object, not instances of different objects of the same object category.

Given the different visual properties observed in the views from toddlers and parents, we next examined whether the special properties of the toddler data can be used to create better object recognition models. To do this, we trained a set of VGG16 networks on only the child data, and independently trained another set on only the parent data, following the methodology and parameters detailed in Section 3.5. Figure 4b summarizes the object recognition accuracy of these models on the clean test set, showing that the toddler models indeed perform much better than the parent-trained models across all training conditions (i.e. using different-sized crops around the gaze center as input). We also investigated the effect of the simulated acuity by repeating all experiments without blurring images. Our results show that blurring was beneficial for learning only when objects were small and views were cluttered (e.g., for adults at 60°/70°), suggesting that foveated vision can help learning by zooming into the area of focus in cluttered views, but can also hinder learning by lowering the overall resolution of the target object. In both cases, either with or without blurring, the main result (that toddler data lead to better model performance) remains true. We note here that similar findings have been reported previously in [3]. However, the previous study was based on a smaller number of participants and did not include eye tracking, but instead trained models with all visible objects in the first-person scene, a less realistic approach. In contrast, the present study used only visually attended objects with moment-by-moment acuity simulations, which is a much closer approximation of the visual data collected by toddler learners and thus allows us to look deeper into what properties of toddler data lead to better object recognition.

## 4.2 Quality in Object Size

The results in the last section suggest that there are special properties of toddler training data that can lead to better visual learning. Using only toddler data we now investigate these properties more closely, starting with the size of attended objects relative to the field of view. To test the effect of size, we first randomly sampled 10,000 frames for which we have object bounding boxes (as detected by YOLO) for the attended object. We approximated the object size with the size of the bounding box. We created two training sets of 5,000 frames each, such that one contained objects smaller than the median and the other contained the larger objects. The median object size was about 10% of the field

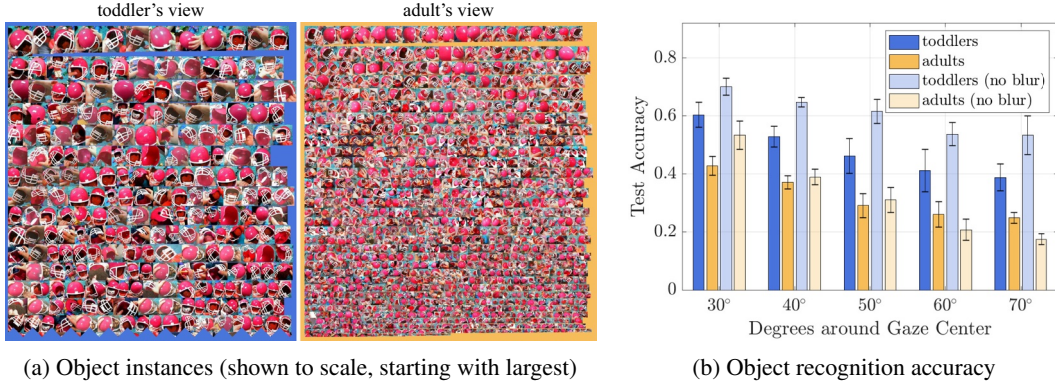

| | |
|---|---|
| (a) Object instances (shown to scale, starting with largest) | (b) Object recognition accuracy |

Figure 4: Comparison of egocentric training data collected by toddlers and parents. (a): Image crops of one example object captured by toddlers and parents are visually very different, in terms of the object size and diversity of views. (b): CNNs trained on the data collected by toddlers significantly outperform those trained with parent data, when tested on a third independent test set.

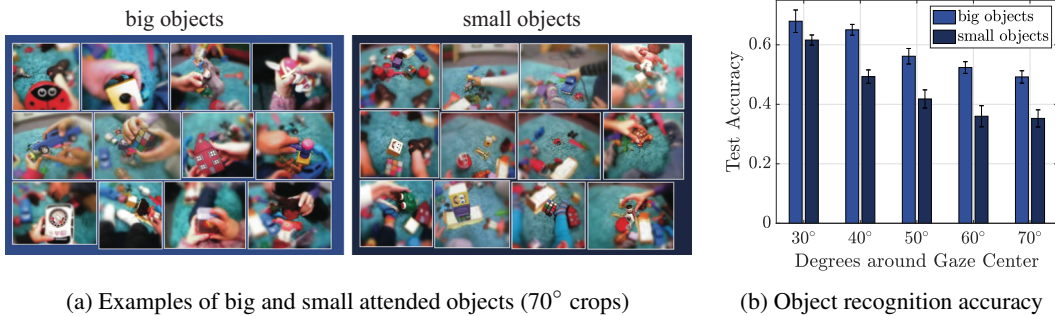

| | |
|---|---|
| (a) Examples of big and small attended objects (70° crops) | (b) Object recognition accuracy |

Figure 5: Effect of object size on the quality of the training data. (a): We split the child training dataset into small and large instances. (b): The subset consisting of large instances led to an object model that performed significantly better on our test dataset.

of view, and the instances in the two subsets had mean sizes of 6.8% and 19.6% of the field of view, respectively. Note that the YOLO boxes were used only to split the instances into subsets, and the training samples themselves were still based on fixed-size crops around the center of gaze. Figure 5a shows examples (70° crops) of training instances with small and large attended objects.

We then trained separate but identical VGG16 networks on the small and large objects collected from the child head cameras. As shown in Figure 5b, the model trained with the large objects achieved significantly better accuracy on the test dataset than that trained with the small objects. Given that the object instances used in training were all resized to $224 \times 224$ pixels before being fed into the CNNs, the results here were not due to a direct size effect, but presumably because larger objects created higher-fidelity instances after being resized, and this better image quality led to better learning.

## 4.3 Quality in object variability

As shown in Section 4.1, another visual property in toddler data that may contribute to successful object learning is the distribution of object views. Clustering in training data could be a useful property, as it would allow a learning system to detect a central prototype of an object category. Variability among examples could also help by encouraging generalization. As shown in Figure 3b, the distribution of toddler data seems to contain both properties, with a large number of highly similar instances but also a tail of diverse ones. Our working hypothesis is that the specific combination of properties encoded in toddler data may be key for children's successful object learning.

To test this idea, we created three training subsets with the same amount of training data in each: (1) a set of similar objects containing object instances that shared similar appearance; (2) a diverse set containing object instances that were not similar but had different appearances; and (3) a random

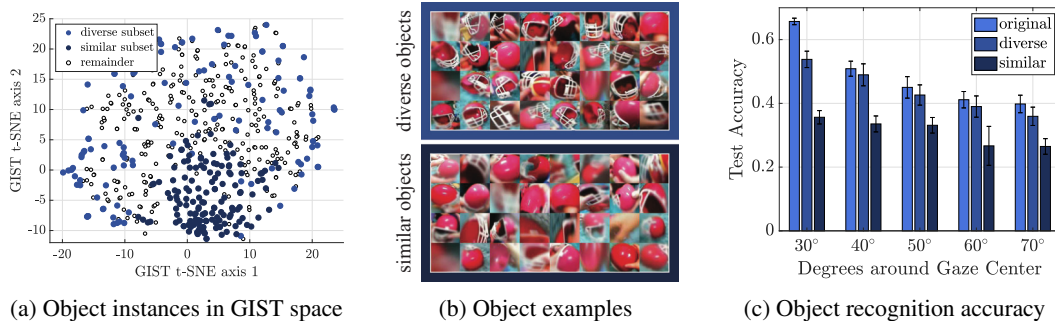

(a) Object instances in GIST space      (b) Object examples      (c) Object recognition accuracy

Figure 6: We study the effect of visual diversity by creating two subsets from the toddler data, one homogeneous and one diverse. (a) Visualization of the similar and diverse object subsets for one object in GIST space. (b) Sample instances from the both subsets. (c) The diverse subset led to a better model than the similar one, but the original dataset having a mixture of the two performed best.

subset of the original set that represents the true object-view distribution generated by toddlers. In particular, we first randomly sampled 10,000 frames for which we had YOLO boxes for the attended object. We cropped out each object using its box and then computed its GIST [31] feature. Using the similarity measure based on GIST, we created a "similar" set consisting of the 25% of instances with (approximately) the minimum total pairwise distance. We also created a "diverse" set containing the 25% of all instances that (approximately) maximized the total distance between all pairs of instances. Finding these subsets involves solving hard computational problems (e.g., the max-sum diversification problem [4]). We use a greedy approximation in which we start with the pair of instances having the largest (or smallest) distance, and then iteratively add the next instance with the largest (or smallest) distance from the centroid of the points selected so far. Figure 6 shows sample images from both sets, as well as a visualization of their distribution in GIST space (projected into two dimensions using t-SNE [18]).

Training the same model using these three subsets, we found that the diverse set outperformed the similar set on our test set, as shown in Figure 6c. This result suggests that seeing different examples helps the model to generalize. More interestingly, the original set outperformed both the similar and diverse sets. Our results suggest that the data created by toddlers, which consists of a mix of both similar and dissimilar instances, is a unique combination of clustering and variability that may be optimal for object recognition.

## 5   Conclusion

Deep learning models have made remarkable progress in matching humans' visual abilities. However, those models rely on large quantities of supervised training data to achieve superior performance. Inspired by toddler learning, the present paper focused on quality of data to understand which fundamental properties in visual training data can lead to successful learning in visual object recognition — one of the most challenging tasks in computer vision and machine learning. Towards this goal, we have conducted a series of simulations which systematically examined how different properties of training data lead to different learning outcomes. We found that image data from toddlers' egocentric views contains unique properties and distributions that are critical for successful learning. This is the first study that has applied deep learning models as formal models to understand visual object recognition in young children, and we believe this methodology can be informative not only for studies of machine learning but also of human learning. Our findings suggest that in everyday toy play, toddlers create their own data with useful properties for learning. While their internal information processing capabilities may not be as sophisticated as those of adults, they have high-quality training data to solve hard learning problems.

More generally, our work suggests that deep learning models may not have to rely on large quantities of training data to reach good performance, but that a smaller number of carefully-selected, high-quality examples may be sufficient. Critically, developing a systematic way to link properties of training data with the learning mechanisms used to process data could eventually allow us to find more efficient ways to train machine learning models.

## Acknowledgments

This work was supported by the National Science Foundation (CAREER IIS-1253549) and the National Institutes of Health (R01 HD074601, R01 HD093792), as well as the IU Office of the Vice Provost for Research, the College of Arts and Sciences, and the School of Informatics, Computing, and Engineering through the Emerging Areas of Research Project "Learning: Brains, Machines, and Children." We would like to thank Drew Abney, Esther Chen, Steven Elmlinger, Seth Foster, Lauren Slone, Catalina Suarez, Charlene Tay, and Yayun Zhang for helping with the collection of the first-person toy play dataset.

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
