[Reviews · NeurIPS 2018]

Reviewer 1



The goal of the paper is to “data mine” records of toddlers’ and their mothers’ fixations while playing with a set of 24 toys in order to observe what might be good training data for a deep network, given a fixed training budget. The idea is that the toddler is the best visual learning system we know, and so the data that toddlers learn from should give us a clue about what data is appropriate for deep learning. They take fixation records extracted from toddlers (16-24 mo old) and their mothers collected via scene cameras and eye tracking to examine the data distribution of infants’ visual input or mothers’ visual input. This study clearly falls under the cognitive science umbrella at NIPS, although they try to make it about deep learning. For example, if they only cared about deep learning, they would not use a retinal filter. First, they manually collect data recording what toys the infants and mothers are fixating on (ignoring other fixations). This must have been a monumental task, as they collected 200 minutes of video at 30 frames a second, resulting in approximately 720,000 frames total from mother and child. This resulted in a dataset of 258k frames for the toddler and 217k frames for the parent. They report two measurements from this data, the size, and the variety. For size, they use a YOLO network to find bounding boxes around the toys in each frame. The mothers’ data has a narrow distribution of size, with most data around 10% of the frame. The toddler distribution is much wider, with a long tail out to 60 or 70% of the frame. This is, of course, because the toddlers are smaller, and when holding toys, the toys subtend a larger image on their retinas. The second statistic is the variation in the images, which they use Torralba’s GIST representation to measure the distance between the GISTs of each pair of frames within the same category, i.e., computing a measure of within-category variance. The toddler data again has a longer tail than the adult data, indicating that their input, while emphasizing similarly-oriented images, also includes more variation that the adult input. They then use Perry & Geisler’s code to filter the data according to a retinal model, centered on the gaze location. This is instance of how it is unclear whether they are building a model of toddler visual learning, or something appropriate to a deep learning system. If they really only cared about deep learning, this step would be unnecessary. They then use transfer learning (with fine-tuning) on a pre-trained VGG16 network to see which set of data is better for training a network to classify the 24 objects. They test on a set of “clean” images of the 24 objects, which were systematically recorded with 128 views of each object on a black background. One issue here is that they use the test data to stop training, a no-no in computer vision, but they appear to think this is ok because the goal is to figure out which dataset is better for training. I’m not quite sure of the logic there. BTW, the images are cropped at 30 to 70 degrees of visual angle. The toddler data wins, of course, or they wouldn’t be writing this paper. Performance is best on the 30 degree data and worst on the 70 degree data. This is important for their inferences from this data, because they are going to claim that it is the size that matters. However, they never report the size of the images in the test data. Since deep networks are not designed to be size invariant, they gain their size invariance from being trained on multiple sizes of the objects. It seems perfectly reasonable to check whether the size of the toys in the test images are closer to the size of the training data in the 30 degree crops, which would explain why size matters. Unfortunately, they don’t report this potential confound. It could be that the size is smaller than the 30 degree crops, and that the network achieved its size invariance in the pre-training stage, since it was pre-trained on imagenet. They should respond to this criticism in the rebuttal stage. It should be easy to run YOLO on their test images and report the size based on that (just because that was the technique for measuring the size in the eye-tracked frames). The characteristics of the toddler data is that the toys take up a larger field of view than their mothers’ data. They then take all of the data and do a median split on the size of the objects and train new networks on this data, and again, the larger images work better for generalization. They also note that the quality of larger images is better, since the objects are higher resolution. They repeat this experiment with variance instead of size, using the quartile with the least variance vs. the quartile with the most variance. The diverse training set wins, but interestingly, training on the toddler-centric data works better than either, so the authors conclude that the toddlers have a good mix of both. Quality: Strengths: The idea is original, the amount of effort that went into this submission (especially collecting the data!) is incredible. I think the question of the potential relationships between human vision and computer vision are very important. Weaknesses: The confound mentioned above is worrisome. The actual relation to deep learning systems is tenuous. Clarity: The paper is very clearly written Originality: I don’t know of anyone else doing this kind of research. This is highly original. Significance: To the extent that this work is relevant to deep learning, it could be significant, but this reviewer is not convinced. However, the significance to human visual development is very high. This is just the first steps on a long research project, I’m sure, and I look forward to more work from this group. Minor comments: line 92: Plutowski, Cottrell, and White (NIPS, 1993) might be a better reference here, as it is using example selection with neural nets. line 182: cross-entry -> cross-entropy I would reverse the bars in Figures 5b and 6c so that they are in the same order as in Figure 4b. Since large objects go with toddlers, and more variability goes with toddlers, it makes sense to put those first. And in 6c, put the toddler-trained bars before the high-variance bar.

Reviewer 2



This paper explores the effect of data distributions on the performance of object classification systems, by specifically looking at data produced by toddlers. Specifically, the paper hypothesizes that the views of objects gathered by head-mounted cameras on toddlers have a combination of (1) high-quality, close-up views and (2) a large number and diversity of rare views (in contrast to views produced by adults in the same setting). The paper suggests that such high-quality datasets (such as those produced by infant wearing a head-mounted camera) may help with the problem of data efficiency. Overall, I think this is an excellent paper and I think it would be great to have it presented at NIPS. It’s generally a good idea to look at how much different data distributions affect the performance of our algorithms, and it seems like an important piece of the puzzle to understand and quantify the data distributions actually perceived by human learners in comparison to our machine learniners. The choice of examining toddler data in particular seems quite unique and I think is very interesting. In general the experiments are quite nice, though the paper could be strengthened by the addition of a few more (described below). Quality --------- Generally the quality of the paper is quite high, but I think a few more experiments could be performed and a few more details given regarding the datasets and results. I would appreciate it if the training accuracy could be reported as well as the test accuracy. In particular, it would be interesting to know whether the adult data was also more difficult to train on than the child data, or whether the difference only appeared at test time. If the latter is true, that would indeed be a very interesting finding. I would like to see the analyses in Figure 3 performed for at least one standard computer vision dataset, like ImageNet, in order to better understand whether it is closer to the parent distribution or the child distribution. The paper suggests it would be closer to the parent distribution since the photos are taken by adults, but this is an empirical question. An additional experiment would be to control for object size and test for the difference in diversity, by cropping the images so that the object takes approximately the same amount of space in the image in both the child and adult datasets. Then, any difference in performance on the test set would be due to diversity in view, rather than size. If there isn’t a difference in performance, then it suggests the main factor between adults and children is size, rather than diversity in viewpoint. This seems like it would be an important variable to isolate. I think there is one issue in the experiments regarding object size, which has to do with the size of the objects in the test set. Specifically, if the objects in the test set are large, then it could be that by increasing the object size (decreasing the FOV around the gaze center) the training set is being brought closer to the test distribution. In that case, the argument becomes a bit weaker, in that it is more about the child data being closer to the test distribution than the training data. I think the paper should report the histogram of object sizes in the test set and analyze this more closely to see if this is indeed what is happening. The similar/diverse analysis in Section 4.3 felt a bit weaker to me, because it seems to me that the important distinction isn’t just “similar images” vs. “diverse images” but “similar *canonical* images” vs. “diverse images”. However, Figure 6b suggests that the “similar” images aren’t necessarily canonical views. It’s not at all surprising to me that low diversity of non-canonical views leads to worse performance. A far more interesting comparison (with non-obvious results, at least to me) would be to see if low diversity of canonical views also leads to worse performance. A suggestion for how to achieve this would be to choose a small set of images which the authors feel are canonical views as a seed for their procedure for choosing the similar/diverse images, so that the similar images are all clustered around the canonical views instead. Some more minor points: - I would like to know what the overall accuracy of YOLO was, for all frames within a look. “At least one” doesn’t convey much, when the look durations were on average around 50 frames. - Thank you for reporting averages over 10 runs of the network, and error bars too! - Figure 6a might look a bit nicer if a different embedding was used, like t-SNE. Also in Figure 6a, it took me a bit to realize the black points were all the images---I would just clarify this in the caption. - The choice of adding in the blurring according to foveation is interesting, but also makes it a bit harder to compare this dataset to standard image datasets which do not have this information. Since the paper argues it’s not about improving performance so much as exploring the choice of data, this seems like an odd decision to me. Can the authors justify why they included the acuity simulation further, and perhaps also report results without that (maybe in an appendix)? Clarity -------- The paper is very well written and easy to read and understand. I only had a few points of confusion: In the second paragraph of the introduction, the paper states that “more quantity naturally leads to better overall quality”. This confused me until I read the rest of the paragraph, and I think the reason is that the paper uses a different definition of “high-quality” than is typically used in the machine learning literature. Usually, “high-quality” in ML means that there is very little noise/error in the labels, that objects are in a canonical view, etc. After reading the rest of the paragraph I understand what the paper means, but I would suggest this paragraph be rephrased just to avoid confusion over the definition of this term in the first place. Line 78: “a few recent studies have applied deep learning models to data collected by humans” Most real-world machine learning datasets were collected by humans, so I think it would be more accurate to say “nearly all deep learning methods are applied to data collected by humans”. But given the citations, I think this sentence was perhaps meant to convey something different, and more about the intersection of psychological research/data with machine learning? Originality ------------- While egocentric datasets have been previously explored in computer vision (e.g. [1, 2]), I do not believe toddler data has been examined in the same way, nor have there been comparisons between adult and toddler data in the context of machine learning performance (of course this has been extensively explored in the psychological literature). While the paper does not introduce a new model, it does perform a nice set of experiments that better characterize the dimensions that we care about in high-quality data for computer vision. Significance ---------------- Overall, I think this paper could be quite useful to the machine learning and computer vision communities for thinking about the distributions of our datasets from a psychological perspective. One question that I have regarding significance is whether the authors plan to open source the dataset? I think it would be a valuable contribution to the computer vision community and would increase the significance of the paper to release the dataset. However, I realize this might not be possible due to privacy concerns (but perhaps the dataset could be anonymized by blurring out faces, in that case?) [1] http://vision.cs.utexas.edu/projects/egocentric_data/UT_Egocentric_Dataset.html [2] http://imagelab.ing.unimore.it/WESAX/related_datasets.asp Edit after author response: I have read the author response and am quite satisfied with it; I am particularly happy that they have (1) included the results without the acuity blur, (2) included more information about the distribution of ImageNet images, and (3) included information about the size of objects in the test distribution (and I find it very interesting that the results are as good as they are when the size of the objects in the test distribution is 0.75, compared with the size of 0.129 in the children's distribution). I think the community at NIPS would find this paper and its results very interesting and have thus decided to increase my score by one.

Reviewer 3



This manuscript explores how to select training data (given limited budget) for achieving good performance for deep learning systems. Inspired by the learning process of human toddlers, the authors conduct a series of experiments and show that the size, quality, and diversity of the training data plays important role for obtaining better performance of an object classification task. The presented idea is clear, and the the experiments are conducted comprehensively. However I do have several concerns which are outlined below: * The key result presented in this manuscript is qualitative rather than quantitative. I learn that we should use bigger, high quality, and diverse images to train a deep network, but I am less clear on what are the combinations of these factors to get the best performance, given limited budget. Should I use all very big images and a bit diverse, or I can make the dataset pretty diverse but with a portion of small images in it? Since the authors do have a large dataset, I believe some ablation studies with these factor combined are needed to get better insight of how to achieve the best performance. * I am curious to see the generalization power of the key findings. That is to say, besides the toddler dataset the authors use, I would like to see the same idea tested on a public benchmark, say ImageNet. Of course not all 1000 categories are needed, a proof of concept on a small subset with sufficient number of training images should be sufficient. * I would also like to see the effect of blurring the training data. Does the blurring have some effect on the performance compared with using original images? Sometimes the peripheral information also helps visual recognition [1]. Also as the authors suggest that blurring is used to model the contrast sensitivity function of human visual system, has the authors considered using log-polar transformed images, as in Wang & Cottrell [2]? It would also be interesting to see the feature learned in the VGG network, as compared with [2]. * Related work on building models to connect visual object recognition in computer vision and developmental psychology: [2], [3], [4] Evaluation: Quality: Overall quality is good, but we need detailed quantitative analysis in order to better demonstrate the central idea. Clarity: The manuscript is clearly written and is easy to follow. Originality: This is an original work to my best knowledge. Significance: Significance can be better justified by conducting more ablation studies and testing the generalization power. [1] Larson, A. M., & Loschky, L. C. (2009). The contributions of central versus peripheral vision to scene gist recognition. Journal of Vision, 9(10), 6-6. [2] Wang, P., & Cottrell, G. W. (2017). Central and peripheral vision for scene recognition: a neurocomputational modeling exploration. Journal of vision, 17(4), 9-9. [3] Smith, L. B., & Slone, L. K. (2017). A developmental approach to machine learning?. Frontiers in psychology, 8, 2124. [4] Wang, P., & Cottrell, G. (2013). A computational model of the development of hemispheric asymmetry of face processing. In Proceedings of the Annual Meeting of the Cognitive Science Society (Vol. 35, No. 35). ===================post rebuttal=========== On one hand, I agree that the authors have done substantial amount of work for data collection and the work is highly original; on the other hand, I expect the authors to put more emphasis on their core idea, that is, the properties of visual data that lead to better visual object learning (as I mentioned in my first concern). Unfortunately, this issue is not addressed in the rebuttal. So my current take-away on this core idea is - we can use large images with some diversity to build our training set, but I sill do not know how to combine these factors. This is also why I raised my ImageNet concern, as the authors can potentially build a small subset to verify the idea. If the idea can be verified on ImageNet, I believe it will make significant contribution to the computer vision society. However, I do think R2 made a reasonable comment on the amount of work it takes to run such experiment, so it will not be a major point that bothers me. Still, the authors have not satisfactorily addressed my biggest concern (and one concern raise by R2 about the control experiment of size). and I think it impairs the significance of the submission. However, consider the work is highly original and the high quality of the presented experiments, I bump my score to 6.